# HNL Dimer in plasma is a unique and useful biomarker for the monitoring of antibiotic treatment in sepsis

**Per Venge**[1]*, **Christer Peterson**[1,2], **Shengyuan Xu**[1,2], **Anders Larsson**[1], **Joakim Johansson**[3], **Jonas Tydén**[3]

1 Department of Medical Sciences, Uppsala University, Uppsala, Sweden, 2 Diagnostics Development, Uppsala, Sweden, 3 Department of Surgical and Perioperative Sciences, Anaesthesiology and Intensive Care Medicine (Östersund), Umeå University, Umeå, Sweden

* per.venge@medsci.uu.se

**Editor:** Sonu Bhaskar, Global Health Neurology Lab / NSW Brain Clot Bank, NSW Health Pathology / Liverpool Hospital and South West Sydney Local Health District / Neurovascular Imaging Lab, Clinical Sciences Stream, Ingham Institute, AUSTRALIA

## Abstract

### Introduction

Sepsis is a growing problem worldwide and associated with high mortality and morbidity. The early and accurate diagnosis and effective supportive therapy are critical for combating mortality. The aim of the study was to compare the kinetics of four biomarkers in plasma in patients admitted to ICU including sepsis and during antibiotics treatment.

### Methods

The biomarkers evaluated were HBP (Heparin-binding protein), HNL Dimer (Human Neutrophil Lipocalin), HNL Total and PCT (Procalcitonin). Plasma was obtained at admission to ICU and during follow-up at days 2 and 3. Antibiotic treatment was started or reviewed on admission to ICU. The results were compared to SOFA and KDIGO-scores and to survival. 277 patients admitted to ICU were included of which 30% had sepsis. The other groups were categorized as miscellaneous, other medical and trauma.

### Results

The plasma concentrations of all four biomarkers were highly elevated with the highest concentrations in sepsis patients. During the follow-up period HNL Dimer decreased already day 2 and further so day 3 (p<0.00001) in contrast to unchanged concentrations of the other three biomarkers. HNL Total showed the strongest relationships to the clinical scores (p<0.0001) and was by multiples regression analysis independently related to these scores.

### Conclusion

Our data supports and confirms our earlier findings of HNL Dimer being a novel and potentially useful clinical tool in antibiotic stewardship in sepsis. HNL Total reflects epithelial cell activity in the body and is an interesting biomarker for the management of organ failure in such patients.

**Data Availability Statement:** All relevant data are within the paper and its Supporting Information files.

**Funding:** The author(s) received no specific funding for this work.

**Competing interests:** Per Venge is a board member and shareholder of P&M Venge AB which owns patent rights on the use of HNL in infectious disease. I am the part owner of the company Diagnostics Development a P&M Venge AB company, which holds the patents relating to the results of this report. The identities of the patents are included in the disclosure section. My ownership does not alter the adherence to PLOS ONE policies on sharing data and materials.

## Introduction

Sepsis is a growing problem worldwide and associated with high mortality and morbidity. The early and accurate diagnosis and effective supportive therapy are critical for combating mortality [1, 2]. However, currently available biomarkers are not sufficiently reliable to act as single diagnostic tools [3]. In consequence, most patients go through a series of tests, thus delaying detection and the set-in of effective therapy. In addition, there is a need for a better monitoring tool to improve the efficiency of the treatment as the biomarkers currently used are not rapid enough to closely follow the development of the condition i.e. the blood biomarkers CRP, White blood cell counts, Procalcitonin (PCT) or Heparin-Binding Protein (HBP) (Azurocidin) [4–12]. One consequence of the inefficient use of antibiotics in sepsis is the development of anti-microbial resistance (AMR). Anti-microbial resistance was identified by the 2013 World Economic Forum as one of the greatest risks globally to human health. Antimicrobial-resistant infections currently claim at least 50,000 lives each year across Europe and the US alone, with many hundreds of thousands more dying in other areas of the world [13].

Our earlier studies showed the superiority of one of our HNL (Human Neutrophil Lipocalin) assays for the distinction between acute infections caused by bacteria or virus [14, 15]. These studies were based on the activation of neutrophils in whole blood and will require the development of novel technologies and be most suitable for the point-of-care applications. However, our recent studies showed that plasma/serum measurement of certain forms of HNL might complement such applications and be used for monitoring successful antibiotics treatment [16]. Thus, HNL exists in several forms in bodily fluids of which the dimeric form originates exclusively from neutrophil granulocytes [12, 17]. The other forms such as monomeric HNL and heteromeric HNL, in complex with MMP9 (Matrix metalloproteinase 9), may originate from both neutrophil and epithelial cells of which epithelial cells seem to be the main source when HNL measured in plasma/serum. We have constructed several assays for the measurement of HNL in various bodily fluids of which one assay uniquely and specifically measures the dimeric, neutrophil originating form.

In one study on sepsis, we showed that the dimeric form of HNL measured in plasma was unique, since the concentrations were rapidly reduced in response to adequate antibiotics treatment in contrast to several other biomarkers such as PCT [16]. This study therefore confirmed the insufficiency of current biomarkers for monitoring successful antibiotic treatment in patients with sepsis admitted to the intensive care unit. Sepsis and septic shock are major healthcare problems, impacting millions of people around the world each year. Early identification and appropriate management improve outcomes but there are no screening tools or biomarkers with adequate performance to be recommended by current guidelines [18].

Our aim of the current study was to investigate further the potential of HNL Dimer as a management tool in sepsis, since the confirmation of such results should support the notion of adding HNL Dimer to the armamentarium of biomarkers of sepsis management. For this purpose, therefore, we have measured in plasma HPB, HNL Dimer, HNL Total and Procalcitonin in a large group of patients admitted to the ICU of which about 30% were diagnosed with sepsis.

## Methods

### Patients

277 patients were admitted to the ICU in Östersund county hospital in Sweden. As detailed in previous publications [19, 20] the patients were separated based on clinical diagnoses into four categories i.e. Sepsis {n = 78, 47 men (median age 72 years, IQ range 61–77 years) and 31

women (median age 74 years, IQ range 64–76 years)}, Miscellaneous disease {n = 79, 52 men (median age 70 years, IQ range 66–77 years) and 27 women (median age 73 years, IQ range 62–80 years)}, Other medical disease{n = 79, 39 men (median age 66 years, IQ range 42–74 years) and 40 women (median age 58 years, IQ range 31–71 years)} and Trauma{n = 32, 27 men (median age 50 years, IQ range 27–67 years) and 5 women (median age 72 years, IQ range 50–87 years)}. Plasma was obtained at admission to ICU and during follow-up at days 2 and 3. Antibiotic treatment was started or reviewed on admission to ICU by an experienced specialist in infectious disease. Patients under the age of 18 years and those transferred from other ICUs were excluded.

## Ethics approval

The study was approved by the regional ethics review board in Linköping, Sweden. Verbal informed consent was given by the patient or next of kin if the patient was not able. Consent was documented on the patient's inclusion sheet. Verbal as opposed to written informed consent was used since many patients were not able to write due to severity of illness. This procedure was approved by the ethics review board due to the observational character of the study. All methods were performed in accordance with the relevant guidelines and regulations. The regional ethics review board of Linköping is the regional section of the Swedish Ethical Review Authority located in Uppsala (Box 2110, SE-75010 Uppsala, Sweden). Inclusion of the patients took place during one year from March 1, 2012 to February 28, 2013. The data were accessed for research purposes 10/10/2022.

Blood samples were collected on admission to the ICU and on the two following days. Samples were drawn from the arterial catheter in ethylenediamine-tetra-acetic acid (EDTA) tubes and spun to plasma, which was stored at -80˚C until analysed.

## Measurements

HBP was analyzed as shown previously [20]. Procalcitonin was analysed by a kit purchased from Thermo-Fisher (Life Technologies Europe BV, Stockholm, Sweden) and HNL Dimer (Mab763/765) and HNL Total (pab/mab765) by ELISA kits from Diagnostics Development, Uppsala, Sweden. All assays performed with CVs of duplicates below 10%.

## Normal plasma concentrations

The concentrations in plasma of healthy persons (n = 144) were for HNL-Total: 35.7 µg/L (IQ 29.6–42.0 µg/L), for HNL Dimer: 3.6 µg/L (IQ 2.5–4.8 µg/L), for PCT 0.045 µg/L (IQ 0.036–0.057 µg/L). The normal concentrations of HBP (n = 8) were 7.28 µg/L (IQ 6.65–8.35 µg/L).

## Statistics

Non-parametric statistics were used throughout the report unless otherwise indicated i.e. Mann-Whitney U for non-paired samples, Wilcoxon´s test for paired samples, Kruskal-Wallis ANOVA for more than two non-paired samples and Friedman Test for more than two paired samples. Log-linear regressions were used to compare biomarkers and SOFA-score. Receiver operating characteristics (ROC) analysis was used to compare diagnostic performances of the biomarkers. A p-value of <0.05 was considered significant. The statistics program MedCalc® Statistical Software version 22.003 (MedCalc Software Ltd, Ostend, Belgium; https://www.medcalc.org; 2023) was used throughout.

## Results

### The four biomarkers in different clinical categories

Figs 1 and 2 show plasma concentrations of the four biomarkers in the patients admitted to ICU and separated into four categories. For all biomarkers differences were seen between the clinical categories as determined by Kruskal-Wallis ANOVA (p = 0.0006—p<0.00001). The highest concentrations were measured in sepsis. For all four biomarkers the concentrations in sepsis were highly and significantly different from the other patient categories, except for plasma concentrations of HBP in the miscellaneous group which were similar to the concentrations in sepsis. The lowest concentrations were found in patients with trauma. The discrimination between sepsis and trauma was evaluated by ROC-analysis and showed equal AUCs of HNL Total and PCT whereas the AUCs of HNL Dimer and P-HBP were significantly smaller as compared to the AUC of P-HNL Total, p = 0.01 and p = 0.04, respectively (Fig 3).

### The four biomarkers in relation to outcome

In Table 1 we show the relation of admission concentrations in all patients and sepsis patients separately to outcome in terms of the SOFA score (Sequential Organ Failure Assessment), the KDIGO score (The Kidney Disease: Improving Global Outcomes) and 30 days survival.

In the evaluation of the whole cohort, all four biomarkers were significantly related to SOFA-score and kidney function as evaluated by the KDIGO-score. Except for PCT, the

**P-HNL Dimer and P-HNL Total in admission samples**

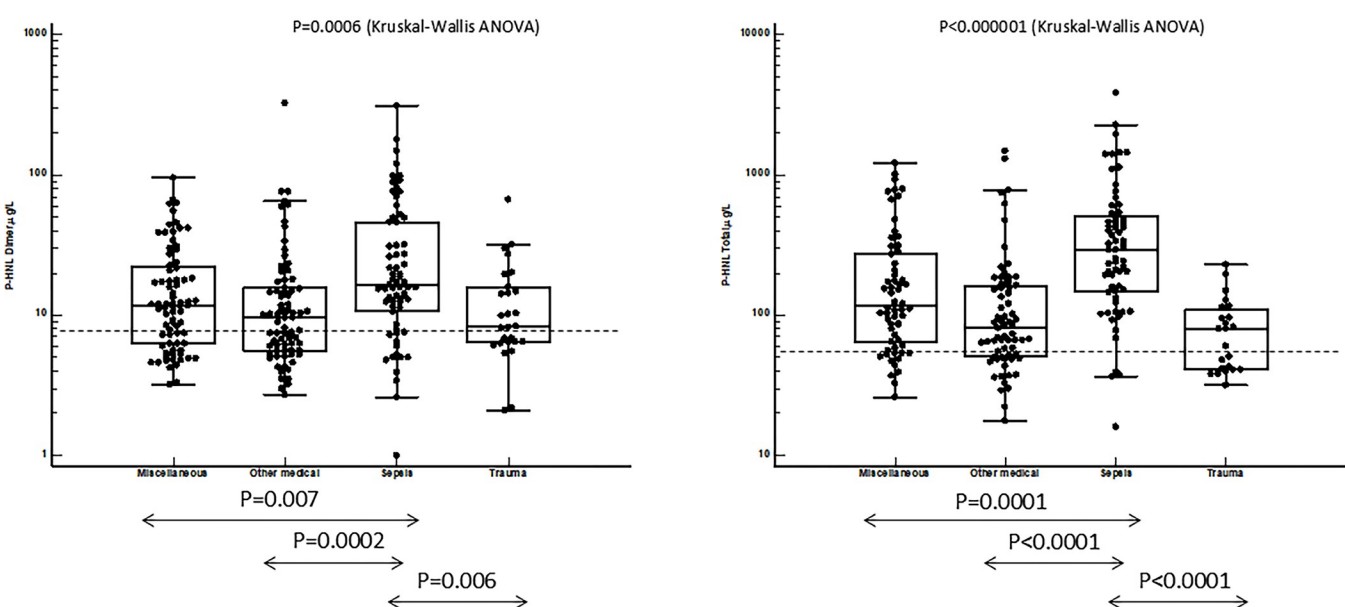

**Fig 1.** Shows the plasma concentrations of P-HNL Dimer (left) and P-HNL Total (right) in the four patient categories: Miscellaneous, Other Medical, Sepsis and Trauma. The statistical differences between sepsis and either of the other three groups of patients are given in the figure. The overall differences in plasma concentrations were calculated by Kruskal-Wallis ANOVA and differences between groups by Mann-Whitney U-test. The broken horizontal lines represent the upper 97.5[th] percentile of healthy subjects.

## HBP and PCT in admission samples

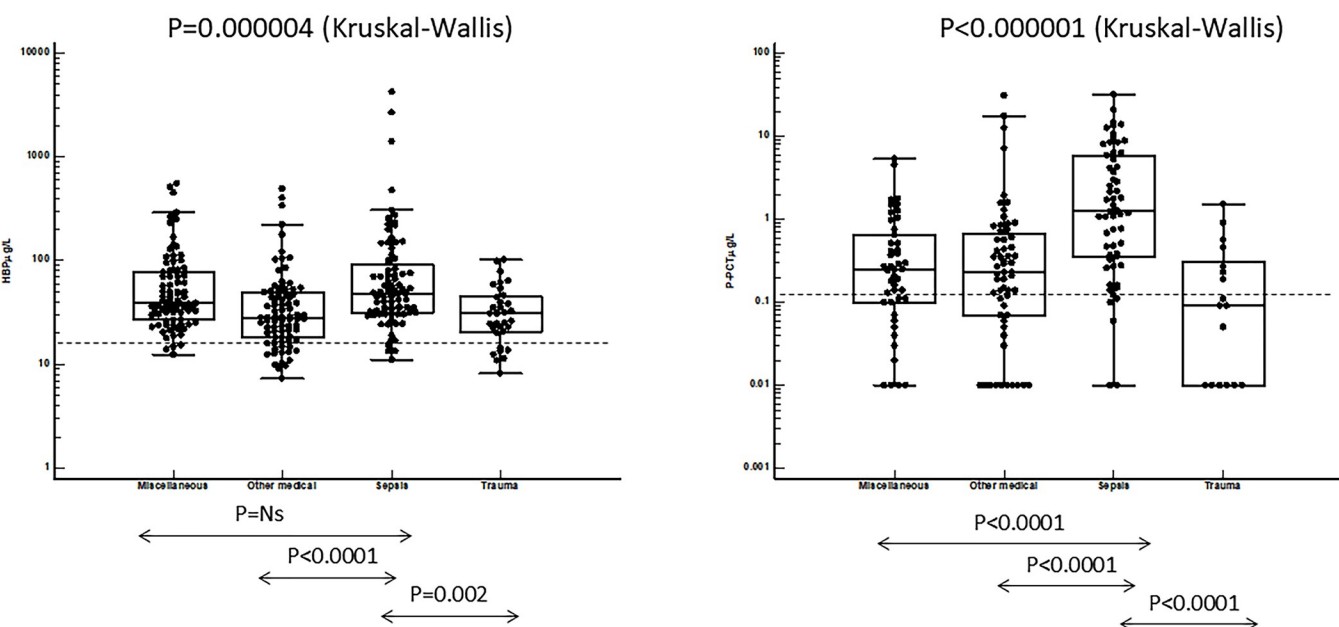

**Fig 2.** Shows the plasma concentrations of P-HBP (Heparin Binding Protein) (left) and P-PCT (Procalcitonin) (right) in the four patient categories: Miscellaneous, Other Medical, Sepsis and Trauma. The statistical differences between sepsis and either of the other three groups of patients are given in the figure. The overall differences in plasma concentrations were calculated by Kruskal-Wallis ANOVA and differences between groups by Mann-Whitney U-test. The broken horizontal lines represent the upper 97.5$^{th}$ percentile of healthy subjects.

biomarkers also showed significant differences between survivors and non-survivors. In the subgroup of patients with sepsis, HNL-Total and PCT showed significant associations to SOFA-score as well as to KDIGO. The latter was also seen for HBP. The HNL Dimer did not show such relations and none of the biomarkers showed significant differences between survivors and non-survivors in the sepsis cohort.

The relationship between SOFA-score and HNL Total and PCT are further illustrated in Fig 4. For both biomarkers highly significant log-linear relationships to SOFA-score were seen (HNL-total r = 0.46 and for PCT r = 0.37). Not shown are the highly significant correlations (p<0.0001) between HPB (r = 0.40) and HNL Dimer (r = 0.32) to SOFA-score, respectively. In a multiple regression analysis including the four biomarkers in addition to sex and age, HNL Total independently related to the SOFA score in the whole cohort (p<0.0001) and in patients with sepsis (p = 0.004).

In Fig 5 the relationships of HNL Total and PCT to KDIGO are illustrated and show close relationships as was also seen for HPB and HNL Dimer in the whole cohort of patients (results not shown). Relations to KDIGO were also seen for HNL Total and PCT in the cohort of patients with sepsis, but only for HNL Total in miscellaneous and other medical disease, but not in patients with trauma in which only few patients had any signs of kidney failure. In a multiple regression analysis including the four biomarkers in addition to sex and age, HNL Total independently related to KDIGO in the whole cohort (p<0.0001) and in patients with sepsis (p<0.0001).

## The discrimination between sepsis and trauma at admission

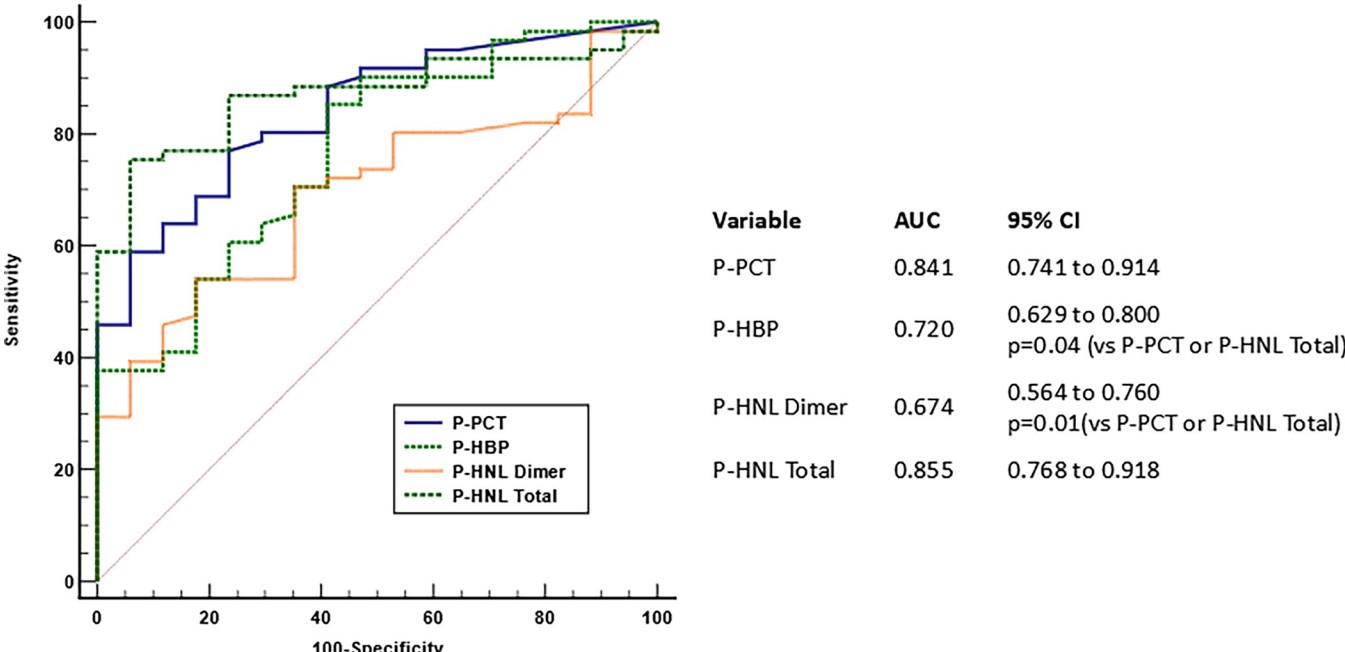

**Fig 3. Shows the discrimination between sepsis and trauma for the four biomarkers by receiver operating characteristics (ROC) curves.** Also shown in the figure are the Area under the curves (AUCs) and their 95% confidence intervals. The AUCs of P-HNL Dimer and P-HBP were significantly smaller than the AUCs of P-HNL Total, p = 0.01and of PCT p = 0.04, respectively.

**Table 1. The four biomarkers in relation to SOFA and KDIGO-scores and to survival.** We show the results for the entire cohort and for sepsis only.

| Biomarker All patients | SOFA score <8 or >7 | p-value | KDIGO (Kruskal-Wallis) | 30 days survival, yes/no | |
|---|---|---|---|---|---|
| HNL Total n = 243 | 48 vs 138 µg/L | P<0.0001 | P<0.0001 | 53 vs 107 µg/L | P<0.0001 |
| HNL Dimer n = 246 | 10.2 vs 16.8 µg/L | P<0.0001 | P = 0.001 | 10.8 vs15.2 µg/L | P = 0.02 |
| HBP n = 277 | 33 vs 51 µg/L | P<0.0001 | P<0.0001 | 34.6 vs 47.7 µg/L | P = 0.0007 |
| PCT n = 180 | 0.27 vs 1.49 µg/L | P = 0.0001 | P<0.0001 | 0.30 vs 0.46 µg/L | Ns |
| **Biomarker Sepsis** | **SOFA score <8 or >7** | **p-value** | **KDIGO (Kruskal-Wallis)** | **30 days survival, yes/no** | |
| HNL Total n = 69 | 91 vs 189 µg/L | P = 0.01 | P = 0.001 | 102 vs 168 µg/L | Ns |
| HNL Dimer n = 64 | 16.2 vs 18.2 µg/L | Ns | Ns | 18.0 vs16.9 µg/L | Ns |
| HBP n = 83 | 43 vs 70 µg/L | Ns | P = 0.03 | 47 vs 60 µg/L | Ns |
| PCT n = 53 | 1.10 vs 3.57 µg/L | P = 0.008 | P = 0.005 | 1.47 vs 1.22 µg/L | Ns |

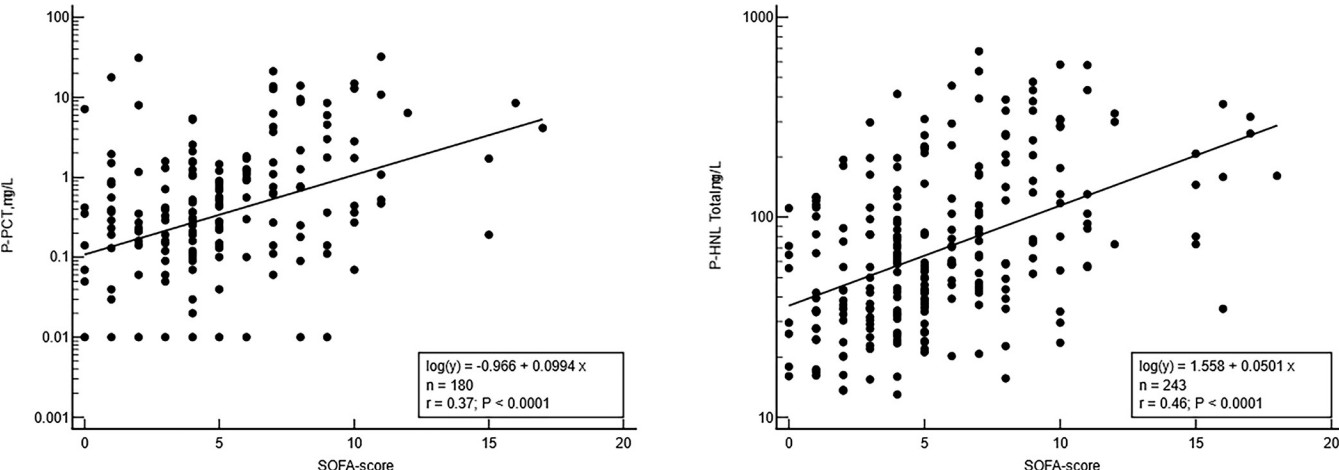

**Fig 4. Shows the log-linear correlations between SOFA-score and the plasma concentrations of P-PCT and P-HNL total including all patients.** The equations of the regression line and r-values are shown in the figures.

## Monitoring of plasma biomarkers in ICU

**Procalcitonin (PCT).** Plasma samples were collected from all patients on three consecutive days. The results of the biomarkers are shown in Figs 6–9. In Fig 6 the results of PCT are shown. Overall, there were no significant differences of PCT concentrations during these three days. Also shown in the figure are the results of patients with sepsis. By Friedman test, in which only paired samples were calculated on, we saw differences between the three days with slightly higher concentrations day 2, but a reduction in concentrations between day 2 and day 3 (p = 0.008, Wilcoxon´s test). However, when we compared the concentrations at the three days including all results on sepsis patients no significant differences were discerned (P = Ns, Mann-Whitney U-test) (Table 2).

**Heparin-binding protein (HBP).** The plasma concentrations of HBP are shown for the whole cohort and for the sepsis cohort separately in Fig 7 and Table 2. No changes in plasma concentrations in either group were seen during the three days follow-up.

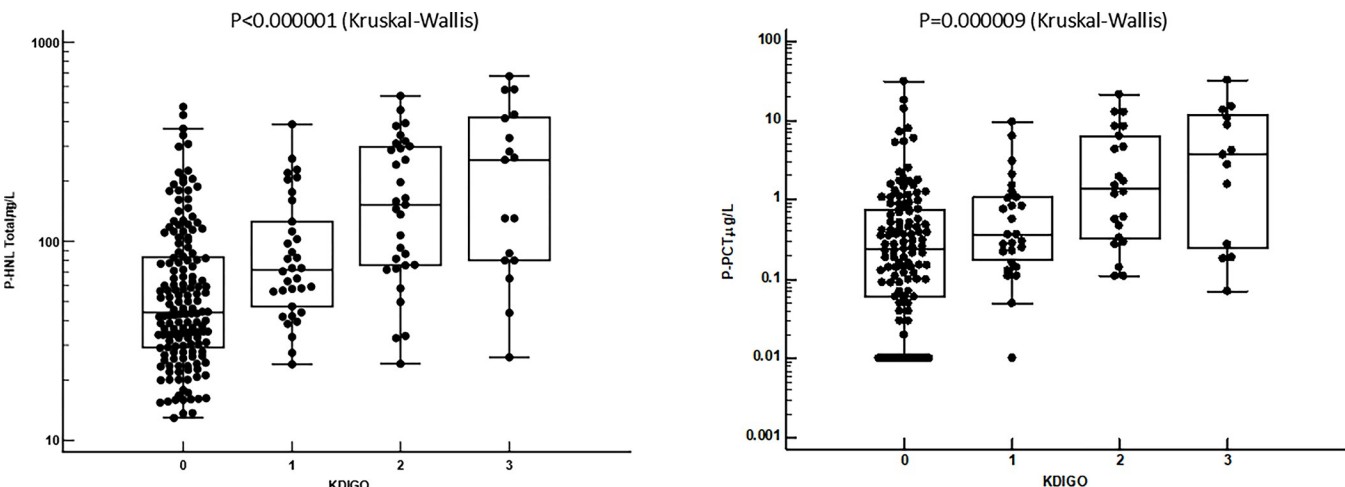

**Fig 5.** Shows the plasma concentrations of P-HNL Total (left) and P-PCT (right) in relation to kidney function as evaluated by the KDIGO score. The overall differences between KDIGO-scores were evaluated by Kruskal-Wallis ANOVA.

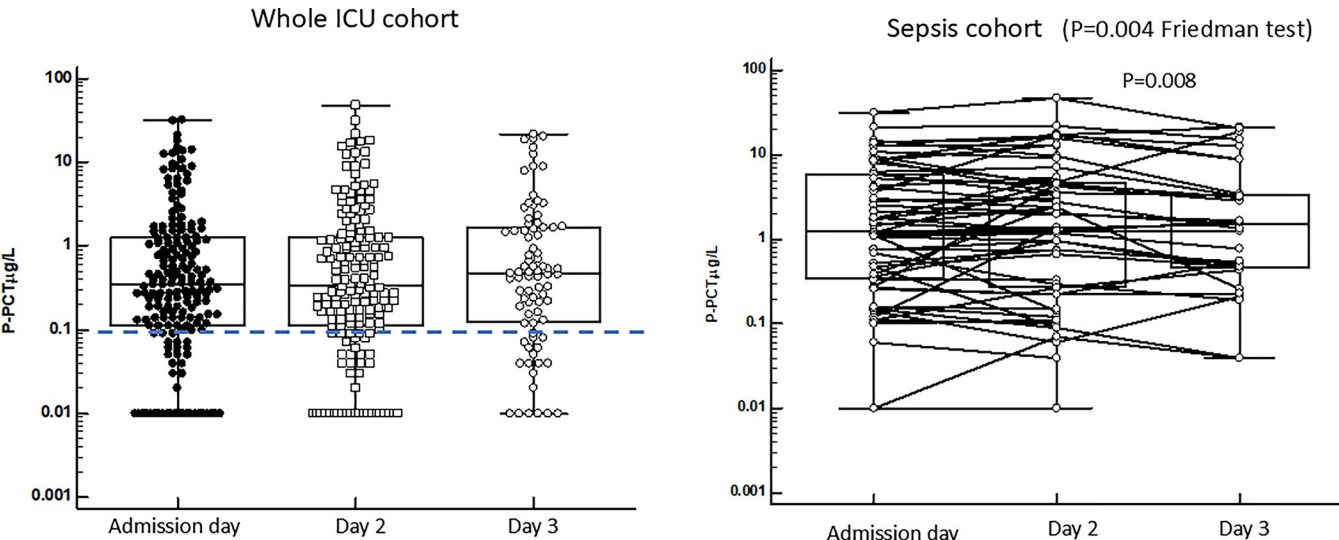

**Fig 6.** Shows the plasma concentrations of PCT at the three days of follow-up for the whole cohort of patients (left). No differences were found in plasma concentrations between the three different days of follow-up. The right panel shows the results in the sepsis cohort. By Friedman test for paired multiple samples we found significant differences between the days (p = 0.0004) and a significant reduction in P-PCT concentrations between days 2 and 3 (p = 0.008, Wilcoxon´s test). The horizontal hatched line indicates 95% percentile concentration of healthy individuals.

**Human neutrophil lipocalin, total (HNL total).** Fig 8 shows the concentrations of HNL Total during the three days. Overall, no differences in plasma concentrations were seen. However, by Friedman test we saw a highly significant change in the sepsis group (P = 0.00008) with a reduction at day 2 (p = 0.0001, Wilcoxon´s test) as compared to day 1 and a further reduction at day 3 (p = 0.001, Wilcoxon´s test). The reductions were 74% (day 2) and 77% (day 3) of the concentrations at admission day (Table 2).

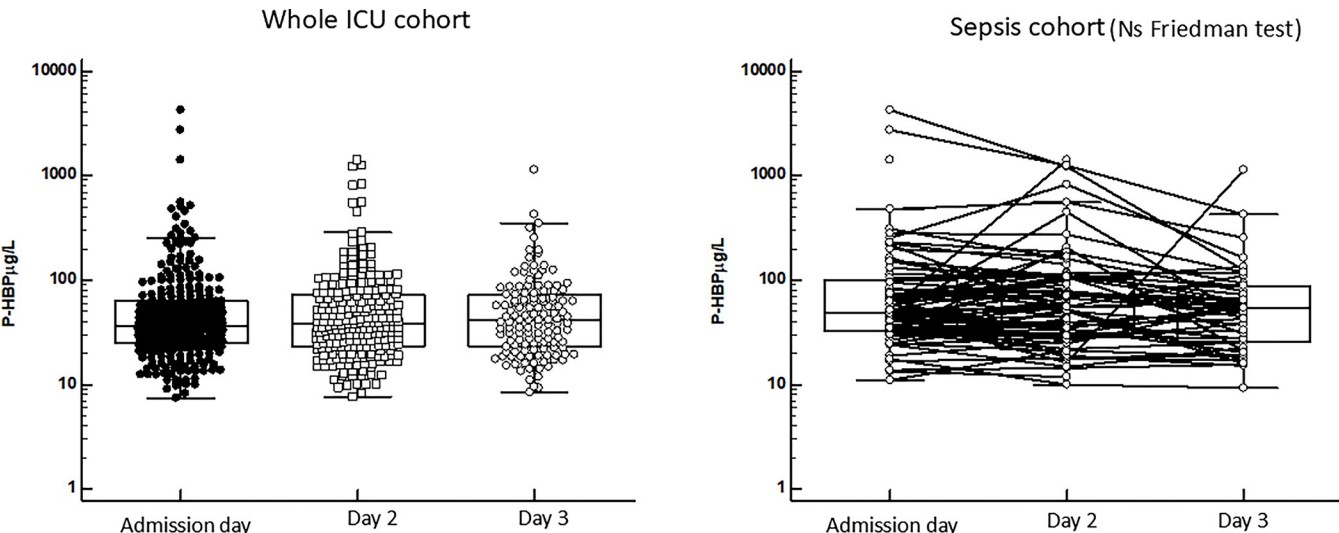

**Fig 7.** Shows the plasma concentrations of HBP at the three days of follow-up for the whole cohort of patients (left). No differences were found in plasma concentrations between the three different days of follow-up. The right panel shows the results in the sepsis cohort. By Friedman test for paired multiple samples no significant differences were found.

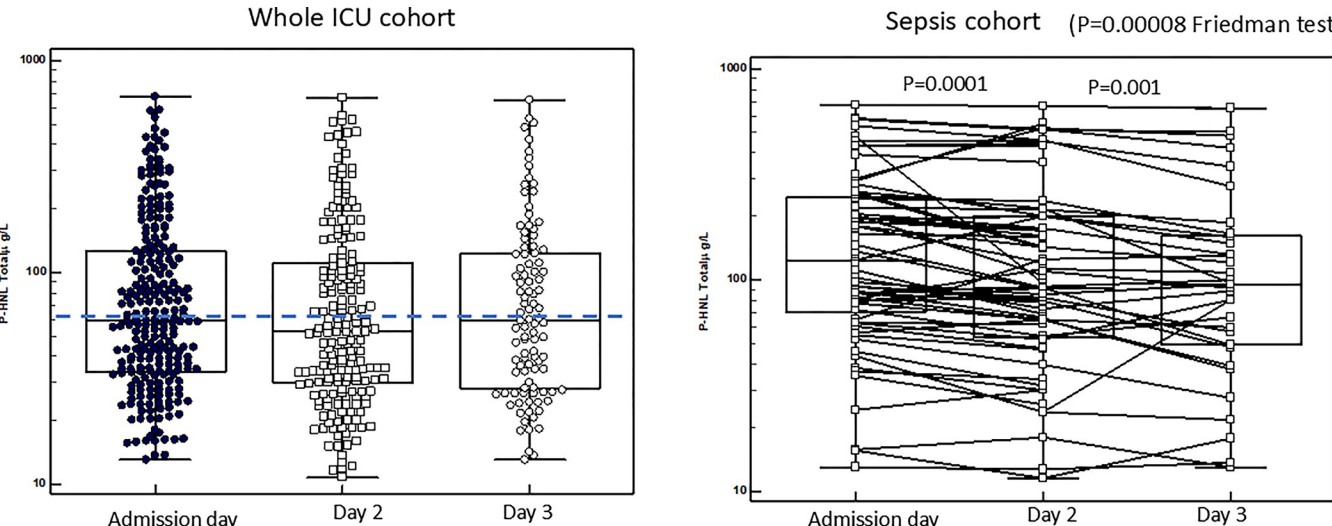

**Fig 8.** Shows the plasma concentrations of P-HNL Total at the three days of follow-up for the whole cohort of patients (left). No differences were found in plasma concentrations between the three different days of follow-up. The right panel shows the results in the sepsis cohort. By Friedman test for paired multiple samples we found significant differences between the days (p = 0.00008) and significant reductions in P-HNL Total concentrations between days 1 and 2 (p = 0.0001, Wilcoxon´s test) and between 2 and 3 (p = 0.001, Wilcoxon´s test). The horizontal hatched line indicates 95% percentile concentration of healthy individuals.

**Human neutrophil lipocalin, dimer (HNL Dimer).** Fig 9 shows the results of plasma concentrations of HNL Dimer. For the whole cohort (median 11.9 µg/L, IQ range 6.3–22 µg/L, n = 246) significant reductions were seen at days 2 (median 8.9 µg/L, IQ range 5.2–15.5 µg/L, n = 205) (p = 0.0005) and 3 (median 7.5 µg/L, IQ range 5.4–12.9 µg/L, n = 102) (p<0.0001, Wilcoxon´s test). In the sepsis group the changes in plasma concentrations from admission day to days 2 and 3 were highly significant (P<0.00001, Friedman test). By paired tests

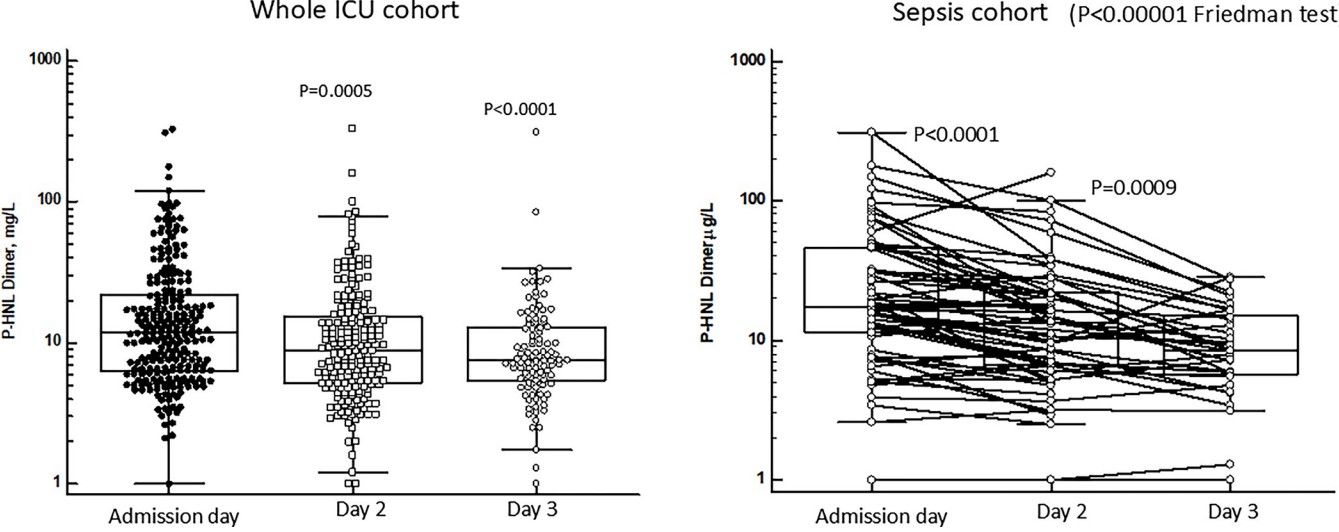

**Fig 9.** Shows the plasma concentrations of P-HNL Dimer at the three days of follow-up for the whole cohort of patients (left). As compared to admission day the concentrations were significantly lower day 2 (p = 0.0005) and 3 (p<0.0001) (Mann-Whitney U test). The right panel shows the results in the sepsis cohort. By Friedman test for paired multiple samples we found significant differences between the days (p<0.00001) and significant reductions in P-HNL Dimer concentrations between days 1 and 2 (p<0.0001, Wilcoxon´s test) and between 2 and 3 (p = 0.0009, Wilcoxon´s test). The horizontal hatched line indicates 95% percentile concentration of healthy individuals.

**Table 2. The four biomarkers are shown for the entire sepsis cohort.** Per cent reductions as compared to admission day are shown. Comparisons between admission day and days 2 and 3, respectively, were calculated by the Mann-Whitney U test and statistical differences indicated in the table.

|  | Admission day | Day 2 | Day 3 |
|---|---|---|---|
| P-HNL Dimer | 17.3 (IQ 11.5–46) µg/L<br>N = 69<br>**100%** | 13.5 (IQ 6.6–22) µg/L<br>N = 62<br>**78% p = 0.01** | 8.5 (IQ 5.8–15) µg/L<br>N = 33<br>**49% p = 0.0001** |
| P-HNL Total | 124 (IQ 70–246) µg/L<br>N = 69<br>**100%** | 92 (IQ 54–200) µg/L<br>N = 61<br>**74%** | 95 (IQ 49–161) µg/L<br>N = 33<br>**77%** |
| P-PCT | 1.27 (IQ 0.34–5.7) µg/L<br>N = 59<br>**100%** | 1.38 (IQ 0.27–4.6) µg/L<br>N = 55<br>**109%** | 1.49 (IQ 0.46–3.4) µg/L<br>N = 31<br>**117%** |
| P-HBP | 49 (IQ 33–99) µg/L<br>N = 83<br>**100%** | 55 (IQ 31–108) µg/L<br>N = 76<br>**112%** | 55 (26–88) µg/L<br>N = 48<br>**112%** |

between admission day (median 17.6 mg/ IQ range 14.2–25.9 µg/L, n = 58) and day 2 (median 13.7 mg/ IQ range 8.8–16.9 µg/L, n = 58) the changes were highly significant (p<0.0001, Wilcoxon´s test) which was also the case for the changes between days 2 and 3 (p = 0.0009, Wilcoxon´s test). By the inclusion of all HNL Dimer results in the sepsis group a similar pattern emerged with a 49% reduction in the plasma concentrations at day 3 (Table 2). The changes in HNL Dimer after antibiotics treatment in relation to 30 mortality were evaluated in the whole ICU cohort and in sepsis separately. For the whole cohort the HNL Dimer concentrations at admission day among those who survived 30 days were 11.9 µg/L (IQ 6.2–23 µg/L) and at day 2, 8.4 µg/L (IQ 5.0–14.3 µg/L), n = 156, p<0.0001 (Wilcoxon´s test for paired results). For those who died during the 30 days of observation the HNL Dimer concentrations at admission day were 13.6 µg/L (IQ 7.2–22 µg/L) and at day 2, 12.0 µg/L (IQ 6.8–21 µg/L), n = 42, p = Ns (Wilcoxon´s test for paired results). For the sepsis cohort the HNL Dimer concentrations at admission day among those who survived 30 days were 17.2 µg/L (IQ 7.7–48 µg/L) and at day 2, 13.7 µg/L (IQ 6.7–22 µg/L), n = 47, p<0.0001 (Wilcoxon´s test for paired results). For those who died during the 30 days of observation the HNL Dimer concentrations at admission day were 15.3µg/L (IQ 11.8–27 µg/L) and at day 2, 13.1 µg/L (IQ 6.4–9 µg/L), n = 8, p = Ns (Wilcoxon´s test for paired results).

**Biomarker comparison, P-HNL Dimer vs PCT, at follow-up.** The differences between P-HNL Dimer and P-PCT during sepsis follow-up are further illustrated in Fig 10. In the figure we show the highly significant reduction of P-HNL Dimer at day 2 after admission to the ICU in contrast to the non-significant changes in P-PCT. It is also seen in the figure that in the six patients with the very high P-HNL Dimer concentrations i.e. > 80 µg/L at admission day the reduction in concentrations was between 5-10-fold at day 2.

## Discussion

The aim of this study was to investigate whether the previous results of successful monitoring of adequate antibiotics treatment with plasma measurements of the dimeric and neutrophil specific form of HNL could be confirmed [16]. The answer to this question is undoubtedly yes, since our findings in the present study showed clearly that HNL dimer is rapidly reduced in sepsis and other ICU-patients after treatment. This contrasted with the findings of the other biomarkers, Heparin-binding protein and Procalcitonin, which showed little change during the three days follow-up after start of treatment (Figs 6 and 7). The concentrations of total HNL also showed some reduction after treatment, but to a much lesser extent than HNL

## Sepsis monitoring

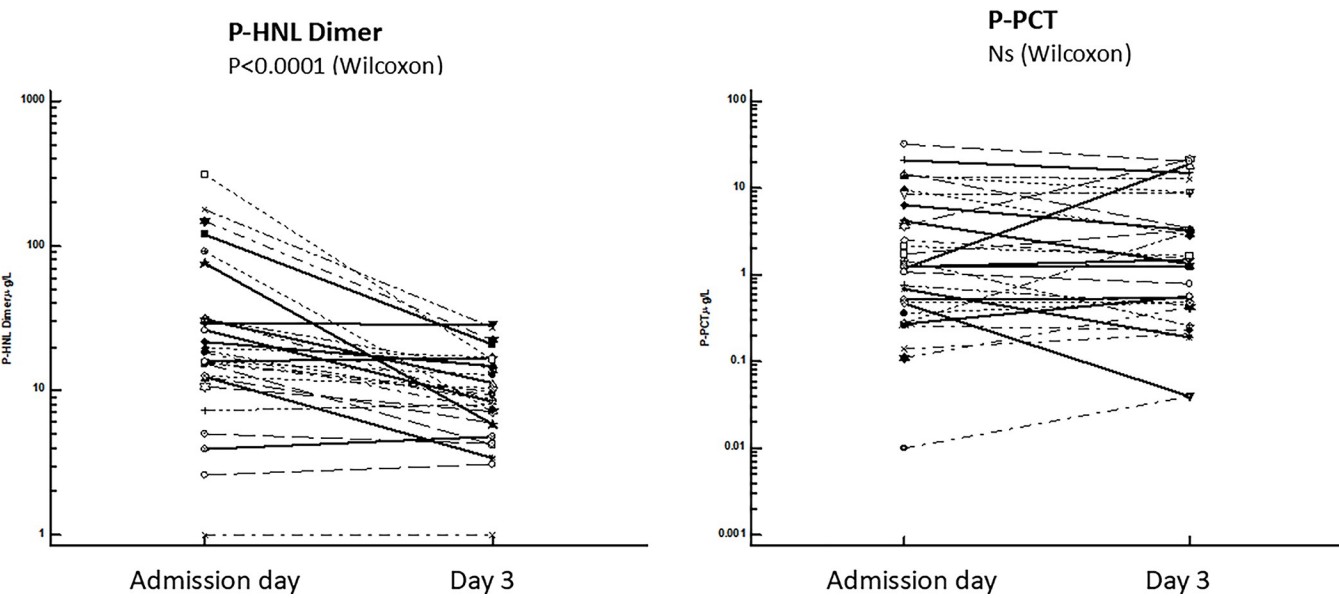

**Fig 10.** Shows for comparison the change in plasma concentrations of P-HNL Dimer (left) and P-PCT (right) between admission day and day 3. The concentrations of P-HNL Dimer decreased significantly (p<0.00001, Wilcoxon´s test) at day 3 in contrast to the concentrations of P-PCT which remained unchanged during the follow-up period.

Dimer. In our previous study on sepsis [16], we found significant reductions of Procalcitonin concentrations day 4 after treatment start and those findings were in keeping with the numerous reports on Procalcitonin kinetics after adequate antibiotics treatment which showed Procalcitonin concentrations to be lowered about 72 hours after such treatment [4–7, 12]. This quality of Procalcitonin has been widely applied in sepsis management in addition to its diagnostic qualities of diagnosing sepsis [21, 22]. However, the relatively slow response to adequate antibiotics treatment shows that the clinical need of early signs of treatment response is not met by this biomarker and the early response by the dimeric form of HNL, therefore, is of particular interest. This response is likely the direct consequence of the reduced bacterial challenge of the neutrophils with the reduction in secretion of dimeric HNL. The fact that the dimeric HNL is uniquely located in neutrophils and that any measurements in bodily fluids of this molecule uniquely reflect neutrophil activities indicates that this molecule belongs to the small group of proteins in the body with single cell origins such as Cardiac Troponin, Insulin and Eosinophil Peroxidase. The differences in kinetics and appearance of the four biomarkers after start of antibiotics may be explained by the fact that HNL Dimer is located preformed in the secondary granules of the neutrophils and readily released upon stimulation of the neutrophil. Other biomarkers such as PCT are inducible and released after de novo synthesis by many different cells after their stimulation by a variety of mechanisms and are therefore expected to appear later in the circulations than preformed biomarkers [23].

The diagnostic qualities in sepsis of HNL Dimer were inferior to the other three biomarkers. These qualities were similar to previous findings and most likely reflect the fact that sepsis is a complex disease involving several organs and organ failure. Thus, the HNL Dimer did not show any relationship to clinical signs reflecting organ failure such as SOFA and KDIGO-

scores in the sepsis cohort in contrast to HNL Total which showed very strong associations to both SOFA and KDIGO-scores. In the multiple regression analysis including all four biomarkers, age and sex, HNL Total was independently related to both SOFA-score and to KDIGO-score both in the total cohort and in the sepsis cohort, which confirms our previous results in Sars-CoV-2 infections [24]. Thus, HNL Total concentrations in plasma above all reflect epithelial cell activities such as tubular secretion in the kidney, but also the secretion by lung epithelial cells.

The other neutrophil biomarker HBP showed no relation to SOFA-score, but a weak and significant relation to KDIGO. However, in the sepsis group none of the biomarkers predicted the outcome as to survival. This was very much contrasted by the findings with the whole patient cohort in which all biomarkers were strong predictors of survival with the sole exception of Procalcitonin. Earlier studies showed that Procalcitonin concentrations predicted outcome as to survival several days after start of treatment since the non-survivors remained with high Procalcitonin concentrations [4, 5]. Additional calculations on day 2 and day 3 results in our patient cohort did not support this notion.

This study was retrospective and not designed to monitor adequate antibiotics treatment, which is a limitation of the study. However, the results showed clearly that the neutrophil originating HNL Dimer dropped after admission to the ICU as a possible sign of adequate antibiotics treatment with less secretion of HNL Dimer because of successful eradication of the bacterial challenge. This notion was supported by the fact that the drop in HNL Dimer concentrations was not observed in patients who died during the observation period suggesting inadequate treatment. This observation was very clear in the whole cohort but could not be properly evaluated in the sepsis group because of low numbers of patients who died in the sepsis group. Thus, further and future studies should be conducted to clarify this point.

## Conclusions

A cornerstone of successful treatment of sepsis is appropriate antibiotic treatment. A biomarker indicating treatment failure (i.e persisting high levels of HNL) could be helpful prompting new cultures and switching antibiotics and thereby possibly improving outcomes.

We conclude from this study that we have confirmed the potential use of HNL Dimer as a rapid responder to antibiotics treatment and most noteworthy in sepsis, since the reduction was seen several days before the reduction in the other two biomarkers, Heparin-binding protein and Procalcitonin. This makes HNL Dimer an interesting candidate biomarker to be included in the diagnostic armamentarium of sepsis and other infectious diseases. HNL Dimer will be launched in short and the cost-effectiveness of this biomarker should be similar to the other biomarkers discussed in this report. We also confirmed that the measurement of HNL Total very strongly reflects organ failure in sepsis and should therefore also be considered as a tool in the diagnosis and management of sepsis and other infectious disease involving organ failure such as kidney failure.

## Supporting information

**S1 File.**
(XLS)

## Author Contributions

**Conceptualization:** Per Venge.

**Investigation:** Per Venge, Christer Peterson, Shengyuan Xu, Joakim Johansson, Jonas Tydén.

**Methodology:** Christer Peterson, Shengyuan Xu, Anders Larsson.

**Writing – original draft:** Per Venge.

**Writing – review & editing:** Anders Larsson, Joakim Johansson, Jonas Tydén.

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
