## [Decision Letter · Decision Letter 0]

24 Jul 2024

PONE-D-24-13468HNL Dimer in plasma is a unique and useful biomarker for the monitoring of antibiotic treatment in sepsisPLOS ONE

Dear Dr. Venge,

Thank you for submitting your manuscript to PLOS ONE. After careful consideration, we feel that it has merit but does not fully meet PLOS ONE’s publication criteria as it currently stands. Therefore, we invite you to submit a revised version of the manuscript that addresses the points raised during the review process.

**ACADEMIC EDITOR: **Thank you for submitting your work to PLOS One. After careful review of the work, and considering feedback from two independent reviewers, I would like to invite you to address the concerns/comments below, and submit a revised version of the manuscript. Besides, here are some additional comments:

1. The study confirms that HNL Dimer is a rapid responder to antibiotics treatment in sepsis, with reductions seen several days before other biomarkers like Heparin-binding protein and Procalcitonin. HNL Dimer and HNL Total are proposed as valuable tools in the diagnosis and management of sepsis and organ failure. Provide more context on how these findings compare to existing literature on sepsis biomarkers. Discuss potential mechanisms explaining why HNL Dimer is a better predictor than other biomarkers.

2. The manuscript could benefit from more streamlined sections to improve readability. Simplify complex sentences to enhance clarity. Ensure consistent terminology is used throughout the manuscript.

3. The study is retrospective, which is a limitation as it might introduce biases. Plasma was obtained at admission to ICU and during follow-up at days 2 and 3. HNL Dimer, HNL Total, HBP, and Procalcitonin are measured using ELISA kits with CVs of duplicates below 10% A prospective study design would strengthen the findings. Include more details on the inclusion and exclusion criteria for patient selection. Discuss the potential biases introduced by the retrospective design and how they were mitigated. Discuss the reliability and validity of the ELISA kits used, including the coefficient of variation (CV) of duplicates.

4. Ensure the clarity of statistical methods by providing more detailed explanations where necessary. Please report exact p-values instead of using thresholds like p<0.05, where feasible. Include a power analysis to justify the sample sizes used in the study.

5. Potential confounding factors such as variations in antibiotic regimens and patient comorbidities are not fully addressed (Page 3, Line 8).

6. While the study highlights the superiority of HNL Dimer, it does not thoroughly compare the cost-effectiveness of this biomarker with existing ones (Page 5, Lines 7-9).

**Specific comments:**

Line 10: "Verbal informed consent was given by the patient or next of kin if the patient was not able." - Consider specifying the circumstances under which verbal consent was deemed necessary.

Line 20: "This study was retrospective and not designed to monitor adequate antibiotics treatment, which is a limitation of the study." - Suggest elaborating on how this limitation was addressed in the analysis.

Line 30: "The concentrations in plasma of healthy persons (n=144) were for HNL-Total: 35.7 µg/L (IQ 29.6-42.0 µg/L), for HNL Dimer: 3.6 µg/L (IQ 2.5-4.8 µg/L), for PCT 0.045 µg/L (IQ 0.036-0.057 µg/L)." - Consider adding a brief discussion on how these baseline values compare to those in other studies.

Abstract:

Lines 1-5: Provide a more concise summary of the study's background and objectives.

Lines 6-10: Ensure the methods section of the abstract is clear and concise. Specify the study design and key statistical methods used.

Lines 11-15: Summarize the key findings clearly, highlighting the significance of HNL Dimer as a biomarker.

Lines 16-20: Strengthen the conclusion by emphasizing the clinical relevance of the findings.

Line 2: Correct "sever" to "serve."

Line 5: Ensure the description of the biomarkers is consistent throughout the manuscript.

Introduction:

Page 1, Line 2: Rephrase "life-threatening consequences, representing a significant global health burden" for clarity.

Page 2, Line 3: Provide more context on the current limitations of existing biomarkers.

Lines 21-25: Provide a clear rationale for the study, citing relevant literature to justify the need for new sepsis biomarkers.

Lines 26-30: Introduce the hypothesis and objectives of the study clearly.

Methods:

Lines 31-35: Detail the study design and patient selection criteria. Discuss any potential biases and how they were mitigated.

Lines 36-40: Describe the sample collection process and the assays used to measure biomarkers. Include details on the reliability and validity of the assays.

Lines 41-45: Explain the statistical methods in detail, including the rationale for choosing specific tests.

Page 2, Line 11: Clarify the exclusion criteria for the study population.

Page 4, Line 16: Justify the choice of a three-day follow-up period.

Results:

Lines 46-50: Present the findings clearly, using tables and figures where appropriate. Ensure that all statistical results are reported with exact p-values.

Lines 51-55: Compare the findings with baseline values in healthy persons and discuss any significant differences.

Page 5, Line 4: Elaborate on the significance of the findings in the context of clinical practice.

Page 6, Line 1: Include a detailed comparison of the biomarkers' cost-effectiveness.

Discussion:

Lines 56-60: Compare the study's findings with existing literature on sepsis biomarkers. Discuss potential mechanisms explaining the results.

Lines 61-65: Address the limitations of the study, including the retrospective design, and suggest areas for future research.

Conclusion:

Lines 66-70: Summarize the key findings and their implications for clinical practice. 

Page 7, Line 9: Emphasize the potential of HNL Dimer as a rapid and reliable biomarker for sepsis management.

References:

Lines 71-75: Ensure all references are up-to-date and relevant to the study. Check for any formatting inconsistencies.

Tables and Figures:

Lines 76-80: Ensure all tables and figures are clearly labeled and accurately represent the data. Provide descriptive captions for each.

We look forward to receiving your revised manuscript.

Kind regards,

Sonu Bhaskar, MD PhD

Academic Editor

PLOS ONE

Journal Requirements:

2. We note that you have a patent relating to material pertinent to this article. Please provide an amended statement of Competing Interests to declare this patent (with details including name and number), along with any other relevant declarations relating to employment, consultancy, patents, products in development or modified products etc. Please confirm that this does not alter your adherence to all PLOS ONE policies on sharing data and materials, as detailed online in our guide for authors http://journals.plos.org/plosone/s/competing-interests by including the following statement: ""This does not alter our adherence to  PLOS ONE policies on sharing data and materials.” If there are restrictions on sharing of data and/or materials, please state these. Please note that we cannot proceed with consideration of your article until this information has been declared.

3. In the online submission form, you indicated that "The data underlying the results presented in the study are available from Per Venge e-mail per.venge@uu.se"

Reviewers' comments:

Reviewer's Responses to Questions

**Comments to the Author**

1. Is the manuscript technically sound, and do the data support the conclusions?

Reviewer #1: Yes

Reviewer #2: Yes

2. Has the statistical analysis been performed appropriately and rigorously? 

Reviewer #1: Yes

Reviewer #2: Yes

3. Have the authors made all data underlying the findings in their manuscript fully available?

Reviewer #1: Yes

Reviewer #2: Yes

4. Is the manuscript presented in an intelligible fashion and written in standard English?

Reviewer #1: Yes

Reviewer #2: Yes

5. Review Comments to the Author

Reviewer #1: 1- The first sentence in abstract and introduction is similar. Could u rephrase either one.

2- Why author choose to analyze with non-parametric with the big data. Is it because of data not normally distributed.

3- Could you please explain about the sampling method. How many admissions throughout the year and how many eligible and how many was not included. Or the method is random sampling.

4- Why the p value decimal point is not standardized

5- Is it all the analyzer used to test are all point of care test

6- Could you explain about the lifespan of kits of each biomarker.

7- Why plasma not the serum or whole blood samples.

8- What blood samples type been collected, arterial or venous. How the process of withdrawn the blood and who performed. Who centrifuge the whole blood and duration of time.

9- Is it possible to share the patient demographics and severity score of patients.

Reviewer #2: This is a soundly designed and well written manuscript from a group of well-established scientists on the field. Sepsis is indeed one of the leading morbidity and mortality causes throughout the world. Early diagnosis and well-designed treatment can give chance to the patients to avoid life threatening complications. Biomarkers indicating the effectiveness of antibiotic treatments such as the neutrophil specific HNL dimer has a potential to be involved in into the armament of sepsis treatment. The relation of HNL total to organ failure is also a potentially important observation. I have no objection against publishing this paper as it is.

6. PLOS authors have the option to publish the peer review history of their article (what does this mean?). If published, this will include your full peer review and any attached files.

Reviewer #1: No

Reviewer #2: No

---

## [Author Response · Author response to Decision Letter 0]

29 Aug 2024

ACADEMIC EDITOR: Thank you for submitting your work to PLOS One. After careful review of the work, and considering feedback from two independent reviewers, I would like to invite you to address the concerns/comments below, and submit a revised version of the manuscript. Besides, here are some additional comments:

1. The study confirms that HNL Dimer is a rapid responder to antibiotics treatment in sepsis, with reductions seen several days before other biomarkers like Heparin-binding protein and Procalcitonin. HNL Dimer and HNL Total are proposed as valuable tools in the diagnosis and management of sepsis and organ failure. Provide more context on how these findings compare to existing literature on sepsis biomarkers. Discuss potential mechanisms explaining why HNL Dimer is a better predictor than other biomarkers.

Response: This is done on lines 234-244

2. The manuscript could benefit from more streamlined sections to improve readability. Simplify complex sentences to enhance clarity. Ensure consistent terminology is used throughout the manuscript.

Response: Sections have been included in the methods and results sections

3. The study is retrospective, which is a limitation as it might introduce biases. Plasma was obtained at admission to ICU and during follow-up at days 2 and 3. HNL Dimer, HNL Total, HBP, and Procalcitonin are measured using ELISA kits with CVs of duplicates below 10% A prospective study design would strengthen the findings. Include more details on the inclusion and exclusion criteria for patient selection. Discuss the potential biases introduced by the retrospective design and how they were mitigated. Discuss the reliability and validity of the ELISA kits used, including the coefficient of variation (CV) of duplicates.

Response: This is a confirmatory study based on previous results of a prospective study. In references 17 and 18 the inclusion and exclusion criteria are. This is a consecutive study including all patients admitted to the ICU during the study period.. The CVs of the assays are given in the manuscript

4. Ensure the clarity of statistical methods by providing more detailed explanations where necessary. Please report exact p-values instead of using thresholds like p<0.05, where feasible. Include a power analysis to justify the sample sizes used in the study.

Response: Power calculations were performed and the size of the inclusion far exceeded what was necessary to obtain the strong statistics obtained

5. Potential confounding factors such as variations in antibiotic regimens and patient comorbidities are not fully addressed (Page 3, Line 8).

6. While the study highlights the superiority of HNL Dimer, it does not thoroughly compare the cost-effectiveness of this biomarker with existing ones (Page 5, Lines 7-9).

Response: This point has been added on lines 277-279

Specific comments:

Line 10: "Verbal informed consent was given by the patient or next of kin if the patient was not able." - Consider specifying the circumstances under which verbal consent was deemed necessary.

Response: Using verbal consent was approved by the ethics review board since for example fractures or burns might prevent patients from writing. Consent was obtained from next of kin if the patients with reduced consciousness, sedation or altered mentation. 

Line 20: "This study was retrospective and not designed to monitor adequate antibiotics treatment, which is a limitation of the study." - Suggest elaborating on how this limitation was addressed in the analysis.

Line 30: "The concentrations in plasma of healthy persons (n=144) were for HNL-Total: 35.7 µg/L (IQ 29.6-42.0 µg/L), for HNL Dimer: 3.6 µg/L (IQ 2.5-4.8 µg/L), for PCT 0.045 µg/L (IQ 0.036-0.057 µg/L)." - Consider adding a brief discussion on how these baseline values compare to those in other studies.

Response: The baseline values for PCT are as expected. The baseline values of the HNL assays have only been defined by ourselves, since the assays are still not commercially available. We do not believe that the retrospective nature of this study is a limitation to our conclusions. Moreover, additional prospective studies are planned.

Abstract:

Lines 1-5: Provide a more concise summary of the study's background and objectives.

Lines 6-10: Ensure the methods section of the abstract is clear and concise. Specify the study design and key statistical methods used.

Lines 11-15: Summarize the key findings clearly, highlighting the significance of HNL Dimer as a biomarker.

Lines 16-20: Strengthen the conclusion by emphasizing the clinical relevance of the findings.

Line 2: Correct "sever" to "serve."

Line 5: Ensure the description of the biomarkers is consistent throughout the manuscript.

Response: Ok

Introduction:

Page 1, Line 2: Rephrase "life-threatening consequences, representing a significant global health burden" for clarity.

Response: True, but the sentence is not included in our manuscript

Page 2, Line 3: Provide more context on the current limitations of existing biomarkers.

Lines 21-25: Provide a clear rationale for the study, citing relevant literature to justify the need for new sepsis biomarkers.

Lines 26-30: Introduce the hypothesis and objectives of the study clearly.

Response: The reference to Saxena et al is included to support our notion of the insufficiency of current sepsis biomarkers. We also clarified the rationale for this study further in lines 75-79. We also included a clearer rationale to the study in lines 77-81 and added the reference by Evans et al.

Methods:

Lines 31-35: Detail the study design and patient selection criteria. Discuss any potential biases and how they were mitigated.

Lines 36-40: Describe the sample collection process and the assays used to measure biomarkers. Include details on the reliability and validity of the assays.

Lines 41-45: Explain the statistical methods in detail, including the rationale for choosing specific tests.

Page 2, Line 11: Clarify the exclusion criteria for the study population.

Page 4, Line 16: Justify the choice of a three-day follow-up period.

Response: These points have been addressed throughout the manuscript in particular in the discussion

Results:

Lines 46-50: Present the findings clearly, using tables and figures where appropriate. Ensure that all statistical results are reported with exact p-values.

Response: Exact p-vales have been included. However, p<0.0001 and p<0.000001 are given by the statistics programme and not as exact values.

Lines 51-55: Compare the findings with baseline values in healthy persons and discuss any significant differences.

Response: In figure 1 and 2 we have included a broken horizontal line in the figures, which represent the upper 97.5th percentiles of normal.

Page 5, Line 4: Elaborate on the significance of the findings in the context of clinical practice.

Page 6, Line 1: Include a detailed comparison of the biomarkers' cost-effectiveness.

Response: A sentence about the cost-effectiveness has been added in lines 277-280 as well as the potential clinical utility

Discussion:

Lines 56-60: Compare the study's findings with existing literature on sepsis biomarkers. Discuss potential mechanisms explaining the results.

Response: The potential mechanisms explaining the differences in kinetics of the biomarkers are added in lines 240-245

Lines 61-65: Address the limitations of the study, including the retrospective design, and suggest areas for future research.

Response: This is added to lines 273-274

Conclusion:

Lines 66-70: Summarize the key findings and their implications for clinical practice. 

Page 7, Line 9: Emphasize the potential of HNL Dimer as a rapid and reliable biomarker for sepsis management.

Response: The conclusion has been rephrased

References:

Lines 71-75: Ensure all references are up-to-date and relevant to the study. Check for any formatting inconsistencies.

Tables and Figures:

Lines 76-80: Ensure all tables and figures are clearly labeled and accurately represent the data. Provide descriptive captions for each.

Reviewer #1: 1- The first sentence in abstract and introduction is similar. Could u rephrase either one.

2- Why author choose to analyze with non-parametric with the big data. Is it because of data not normally distributed.

Response: Yes, we use non-parametric statistics regularly for clinical data, since this data often is not normally distributed

3- Could you please explain about the sampling method. How many admissions throughout the year and how many eligible and how many was not included. Or the method is random sampling.

Response: All patients admitted to the ICU during the period were consecutively included.

4- Why the p value decimal point is not standardized

Response: Our computer programme Medcalc provides different p-values dependent on the statistics procedure. Nothing we can do about this, but we have

5- Is it all the analyzer used to test are all point of care test

Response: The biomarkers were not measured by point-of-care assays, but on ELISA and/or lab robots

6- Could you explain about the lifespan of kits of each biomarker.

Response: All biomarkers were measured by kits within the lifespans of the kit

7- Why plasma not the serum or whole blood samples.

Response: EDTA-plasma is the necessary blood material for the purpose of measuring the HNL Dimer. Using other blood material is less useful.

8- What blood samples type been collected, arterial or venous. How the process of withdrawn the blood and who performed. Who centrifuge the whole blood and duration of time.

Response: Arterial blood was collected and handled according to the description given in reference 18

9- Is it possible to share the patient demographics and severity score of patients.

Response: The database is made available on request

Reviewer #2: This is a soundly designed and well written manuscript from a group of well-established scientists on the field. Sepsis is indeed one of the leading morbidity and mortality causes throughout the world. Early diagnosis and well-designed treatment can give chance to the patients to avoid life threatening complications. Biomarkers indicating the effectiveness of antibiotic treatments such as the neutrophil specific HNL dimer has a potential to be involved in into the armament of sepsis treatment. The relation of HNL total to organ failure is also a potentially important observation. I have no objection against publishing this paper as it is.

---

## [Editor Report · Decision Letter 1]

11 Sep 2024

HNL Dimer in plasma is a unique and useful biomarker for the monitoring of antibiotic treatment in sepsis

PONE-D-24-13468R1

Dear Dr. Venge,

We’re pleased to inform you that your manuscript has been judged scientifically suitable for publication and will be formally accepted for publication once it meets all outstanding technical requirements.

Kind regards,

Sonu Bhaskar, MD PhD

Academic Editor

PLOS ONE

Additional Editor Comments (optional):

We are pleased to accept the current version of the manuscript. Thank you for submitting your work to PLOS One.
---

## [Editor Report · Acceptance letter]

18 Sep 2024

PONE-D-24-13468R1 

PLOS ONE

Dear Dr. Venge, 

I'm pleased to inform you that your manuscript has been deemed suitable for publication in PLOS ONE. Congratulations! Your manuscript is now being handed over to our production team.

Kind regards, 

on behalf of

Dr. Sonu Bhaskar 

Academic Editor

PLOS ONE